# POLICY CONTRASTIVE IMITATION LEARNING

## ABSTRACT

Adversarial imitation learning (AIL) is a popular method that has recently achieved much success. However, the performance of AIL is still unsatisfactory on the more challenging tasks. We find that one of the major reasons is due to the low quality of AIL discriminator representation. Since the AIL discriminator is trained via binary classification that does not necessarily discriminate the policy from the expert in a meaningful way, the resulting reward might not be meaningful either. We propose a new method called Policy Contrastive Imitation Learning (PCIL) to resolve this issue. PCIL learns a contrastive representation space by anchoring on different policies and generates a smooth cosine-similarity-based reward. Our proposed representation learning objective can be viewed as a stronger version of the AIL objective and provide a more meaningful comparison between the agent and the policy. From a theoretical perspective, we show the validity of our method using the apprenticeship learning framework. Furthermore, our empirical evaluation on the DeepMind Control suite demonstrates that PCIL can achieve state-of-the-art performance. Finally, qualitative results suggest that PCIL builds a smoother and more meaningful representation space for imitation learning.

## 1 INTRODUCTION

Imitation is one of the fundamental capabilities of an intelligent agent (Hussein et al., 2017). Animals and humans can acquire many skills by mimicking each other (Byrne, 2009). In engineering, imitation learning also enables many robotics applications. One mainstream class of imitation learning algorithms is the adversarial imitation learning (AIL) (Ho & Ermon, 2016). AIL converts the imitation task into a distribution matching problem and proposes to imitate it by training a policy against an adversarial discriminator. AIL has enjoyed great success on many imitation tasks: it achieves superior performance (Ho & Ermon, 2016; Kostrikov et al., 2018), and has been experimentally proven to alleviate some of the distributional drift issue, and can work even without expert actions (Torabi et al., 2018b).However, AIL is hard to train in practice, usually involving careful tuning of discriminator neural network sizes and learning rates (Wang et al., 2017; Kim & Park, 2018; Orsini et al., 2021). The fragility of the discriminator (Peng et al., 2018) not only leads to poor performance but also severely limits the applicability of AIL to a broader range of tasks.

Numerous techniques have been proposed to improve the performance of AIL, such as using regularization and gradient penalties (Fu et al., 2017; Kostrikov et al., 2018; Gulrajani et al., 2017). Some works also propose to use different distribution metrics (e.g., KL divergence, Wasserstein distance) (Xiao et al., 2019) for distribution matching and show some improvements. Though these methods show encouraging results, we notice that they ignore one crucial aspect of the problem: the representation of AIL's discriminator. To be specific, the discriminator in AIL is usually trained with binary classification loss that distinguishes expert transitions from agent transitions. This discriminator is then used to define rewards. However, since the only goal of the discriminator is to distinguish the expert from the agent, it does not necessarily learn a good, smooth representation space that can provide a reasonable comparison between the behavior of two agents.Ideal representations should be able to provide semantically meaningful signals to compare the expert policy and the agent policy.

In this paper, we propose a new algorithm called Policy Contrastive Imitation Learning (PCIL) to achieve this goal. Instead of training with a binary-classification objective, we propose to train a discriminator representation space with the contrastive learning loss. Our method differs from the prior representation learning approach in AIL in that we perform contrastive learning between different

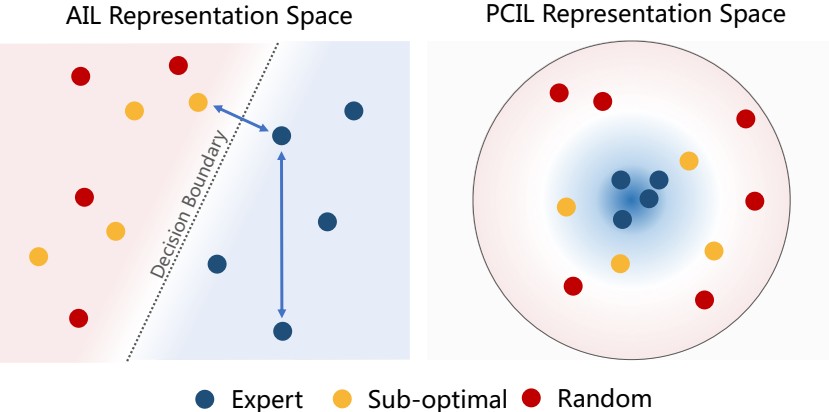

Figure 1: Comparison between the representation space of AIL method and our method. Since the AIL methods use a binary classification objective to distinguish expert and non-expert transitions, the representation space is only required to separate two classes in two disjoint subspaces. So the embedding space is not required to be semantically meaningful enough, e.g. **(Left)** the distance between 2 expert data points may be even longer than the distance between expert data point and sub-optimal non-expert data point. **(Right)** We overcome this limitation by proposing PCIL. Our method enforces the compactness of the expert's representation. This ensures that the learned representation can capture common, robust features of the expert's transitions, which leads to a more meaningful representation space.

policies. More specifically, we push the expert's representation together and pull the agent policy's representation away from them. We define the imitation learning reward via cosine similarity between the policy's and expert's transition.

As is shown in Figure 1, the discriminator (binary classifier) might not have a good representation space: the distance between two expert transitions can be even larger than the distance between an expert transition and an agent transition. This implies that the discriminator may not encode some common features of the expert's behavior and may use non-robust features to compare the behavior. The lack of proper structure in the representation space of traditional discriminators may yield low-quality AIL rewards. To alleviate this issue, we explicitly define a constraint on the representation space, requiring that the distance between expert transitions be smaller than their distance to the agent's transitions. This is a stronger constraint on the discriminator's representation: it is easy to derive a binary classifier from our learned representation. However, the binary classifier's representation space does not necessarily satisfy our objective.

From a theoretical perspective, we show the soundness of PCIL in the apprenticeship learning framework. We also evaluate our method empirically by benchmarking our method on the DeepMind Control Suite (Tassa et al., 2018). Experimental results show that our method is able to achieve state-of-the-art results. Through ablation study and qualitative visualization, we find that our method is more effective than prior representation learning methods and able to provide a better representation space for imitation learning.

In summary, our contributions in this paper are as follows.

1. We point out a new direction to improve the performance of the AIL methods, i.e., going beyond naive binary classification and leveraging more stable and meaningful representation learning algorithms for imitation.

2. We propose an algorithm called Policy Contrast Imitation Learning (PCIL) method that instantiates such an improvement and establishes its connection to apprenticeship learning from a theoretical perspective.

3. We evaluate our method on the DeepMind Control Suite and achieve state-of-the-art performance. Through ablation studies, we highlight its essential difference from previous contrastive learning methods in AIL.

## 2 PRELIMINARIES

### 2.1 NOTATIONS

In this paper, we model the imitation leanring problem as a markov decision process $\mathcal{M} = (\mathcal{S}, \mathcal{A}, p_0(s), p(s' \mid s, a), r(s, a, s'), \gamma)$. Here, $\mathcal{S}$ is the state space. $\mathcal{A}$ is the action space. $p_0(s)$ defines the initial state distribution. $p(s'|s, a)$ defines the transition dynamics. $r(s, a, s')$ is the reward function. $\gamma$ is the discount factor. The goal is to maximize the expected return of the learned policy $\pi$, which is defined by

$$\mathcal{J}(\pi) = \mathbb{E}_{s_0 \sim p_0(s), a_i \sim \pi(\cdot|s_i), s_{i+1} \sim p(\cdot|s_i, a_i)} \left[ \sum_{k=0}^{\infty} \gamma^k r\left(s_k, a_k, s_{k+1}\right) \right].$$

For the imitation learning problem, the algorithm does not have access to the reward function and the transition dynamics. Instead, it is provided with an expert demonstration dataset $\mathcal{D}$ sampled from an expert policy $\pi_E$, which can perform well in $\mathcal{M}$. Here, $\mathcal{D}$ takes the form of $\{(s_i^E, a_i^E)\}$, where $(s_i^E, a_i^E)$ is sampled from $\rho_{\pi_E}$, the stationary state-action visiting distribution of $\pi_E$. The imitation learning algorithm is then required to reproduce the expert's behavior with $\mathcal{D}$.

### 2.2 ADVERSARIAL IMITATION LEARNING

One popular class of the imitation learning algorithm is the adversarial imitation learning (AIL), whose vanilla version is Generative Adversarial Imitation Learning (GAIL) (Ho & Ermon, 2016). The idea of GAIL is to minimize the divergence between $\rho_{\pi_E}$ and $\rho_\pi$. It uses a discriminator $D(s, a)$ to distinguish expert's transitions $(s_i^E, a_i^E) \sim \mathcal{D}$ from the policy transitions $(s, a) \sim \rho_\pi$, which is trained by maximizing the objective

$$\mathcal{L} = \mathbb{E}_{(s_i, a_i) \sim \pi}[\log(D(s_i, a_i))] + \mathbb{E}_{(s_i^E, a_i^E) \sim \mathcal{D}}[\log(1 - D(s_i^E, a_i^E))].$$

To achieve imitation, the agent policy is then required to fool the discriminator, which can only be possible when the policy $\pi$ resembles the expert $\pi_E$. Specifically, GAIL defines an adversarial reward $r(s_t, a_t) = -\log(1 - D(s_t, a_t))$, and trains $\pi$ to maximize the expected return with respect to this reward using on-policy RL algorithms.

## 3 POLICY CONTRASTIVE IMITAITON LEARNING

### 3.1 OVERVIEW

We propose a novel representation-learning-based approach called Policy Contrastive Imitation Learning to improve the AIL reward. The overview of PCIL is illustrated in Figure 2. Our key insight is to learn a policy-contrastive representation space. Unlike the contrastive learning studied in previous AIL literature, the policy-contrastive representation here is obtained by anchoring on policy, leading to meaningful representation that can compare different policies. We will discuss the training of this representation in Section 3.2 and the reward design in Section 3.3. Then, we will show the convergence of our algorithm in Section 3.4.

### 3.2 CONTRASTIVE POLICY REPRESENTATION FOR IMITATION

The vanilla AIL algorithms are based on unconstrained representations and can be very non-robust. One possible approach to handle this problem is to learn a more meaningful representation by contrastive learning. Researchers (He et al., 2020) have found that it can usually learn semantically meaningful representation, leading to better performance on downstream tasks, such as classification. However, though it is effective in the field of supervised learning, prior work (Chen et al., 2021) has found that it does not greatly improve AIL much.

To understand the reason behind this, we first recall that the contrastive learning method learns by drawing the representation of one training sample $\mathbf{x}$ towards a similar positive sample $\mathbf{x}_p$, and pulling it away from a dissimilar, negative sample $\mathbf{x}_n$. Some heuristic rules determine the choice of positive and negative samples: the positive sample is usually defined as a data augmentation of

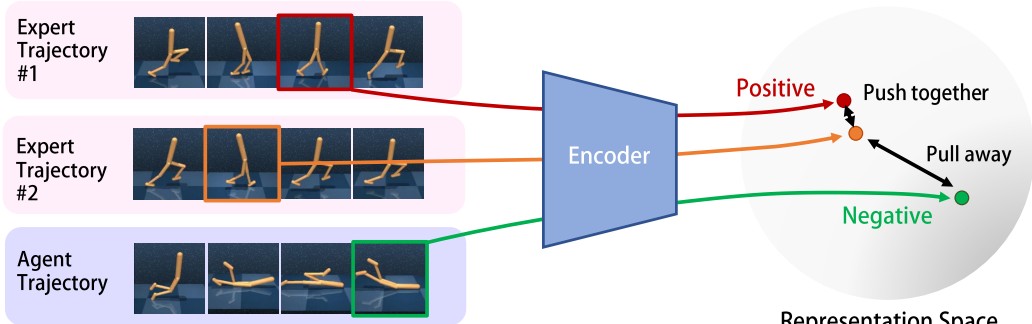

Figure 2: Illustration of our contrastive learning approach. We first select an anchor state (the orange) from the expert trajectory. Then, we select a positive state sample (the red) from another expert trajectory and a negative state sample (the green) from the agent trajectory. We map these selected states to the representation space. Finally, we push the representation of the anchor state and the positive state together and pull the representation of negative samples away from the representation of the anchor state.

$\mathbf{x}$. Therefore, this augmentation decides what should be similar to the representation. Nevertheless, we argue that the definition of similarity in the previous works is not strong enough for AIL. This is because, in AIL, the representation should also be able to let us discern good behavior from the bad behavior. However, the difference between good and bad behavior here can sometimes be very faint. For example, consider a case where a robot misses the exact point to execute a certain action and, as a result, fails to accomplish the task. The good and bad states right before this point may look very similar. More specifically, the difference can simply be minor in a particular physical measurement, like the distance. In this case, the representation should consider this as a semantic component and be sensitive to such a difference to succeed. Unfortunately, the difference between the positive sample and the anchor sample in the previous representation learning methods is usually very large and overwhelms the difference between good and bad states. As a result, the model may not distinguish between good and bad states effectively. Though combining the representation learning objective and the AIL objective may help combat this problem by enforcing a hard distinguishing constraint over contrastive representation, in practice we find this does not work well (Section 4.4).

These observations motivate us to learn a representation that is semantically meaningful and able to distinguish between good and bad states. We find a surprisingly simple yet very effective approach: we can consider the samples drawing from the same policy as the positive samples and the samples from all the other policies as negative samples. In the case of imitation learning, our samples can be naturally divided into two categories, namely expert and non-expert samples. Then given the encoder $\Phi : \mathcal{S} \to \mathbb{S}$ that maps the state to a representation vector in a high-dimensional sphere, we define its infoNCE representation loss function as follows:

$$\mathcal{L} = \mathop{\mathbb{E}}_{\substack{\mathbf{x}_0=(\mathbf{s},\mathbf{a})\sim\mathcal{D}, \\ \mathbf{x}_p=(\mathbf{s}_p,\mathbf{a}_p)\sim\mathcal{D}, \\ \tilde{\mathbf{x}}_i=(\tilde{\mathbf{s}}_i,\tilde{\mathbf{a}}_i)\sim\rho_\pi}} \left[ -\Phi(\mathbf{x}_0)^T\Phi(\mathbf{x}_p) + \log\left( \exp\Phi(\mathbf{x}_0)^T\Phi(\mathbf{x}_p) + \sum_{i=1}^{n} \exp\Phi(\mathbf{x}_0)^T\Phi(\tilde{\mathbf{x}}_i) \right) \right].$$

(1)

Here, $\mathbf{x}_0$ is some state-action pair from the expert transitions; $\mathbf{x}_p$ is some transition from the expert data acting as positives; $\tilde{\mathbf{x}}_i$ is some agent transition working as negatives. In other words, we require $\Phi$ to draw the expert samples towards each other and pull all the policy samples away from the expert samples.

Our proposed objective function is a strictly stronger constraint on the discriminator. The binary classification discriminator can find any hyper-plane that separates the two types of transitions, with no constraint on how the transitions are embedded. However, our objective enforces the pair-wise distance constraint between any triplets. One can derive a binary expert-policy transition classifier from a trained $\Phi$ by computing $\Phi(\mathbf{x}_0)^T\Phi(\mathbf{x}) > t$, where $\mathbf{x}_0$ is any expert transition, $\mathbf{x}$ is the tran-

sition to be classified and $t$ is some threshold. However, on the other hand, the binary classification induced latent space might not satisfy our constraint. As illustrated in Figure 1, the latent space for the binary classification discriminator might have some expert-expert pairs that are even further away than some expert-agent pairs.

Interestingly, our approach echoes the supervised contrastive learning (SCL) (Khosla et al., 2020), which suggests that we consider all the samples in one class as similar, positive samples. Here we also consider all the samples in the expert demonstrations as similar. However, unlike SCL, we do not require the samples from the agents to be similar to each other. This is because the agent transitions are generated from different policies during the training process.

### 3.3 SIMILARITY-BASED IMITATION REWARD

With a representation that can capture the difference between good and bad states, we can then define a reward function to encourage imitation learning. Though using an AIL-style reward with this representation is still possible, we find that a better choice is to use a cosine similarity metric to define the reward. It has several advantages: it is bounded and appears relatively smooth in practice, leading to more stable learning. Concretely, we define:

$$r(\mathbf{x}) = \Phi\left(\mathbf{x}\right)^T \mathbb{E}_{\mathbf{x}_E \sim \mathcal{D}} \Phi\left(\mathbf{x}_E\right). \tag{2}$$

Nevertheless, in practice evaluating the latter expectation can be time-consuming since $\Phi$ is frequently updated. Therefore, we use a random expert sample for the reward calculation. From this reward, we can see that a policy can only obtain high rewards when it frequently visits the expert's distribution. This naturally connects our method to the distribution matching, and we provide the theoretical analysis of our algorithm in the following subsection.

### 3.4 THEORETICAL ANALYSIS

In this part, we show that PCIL can be reduced to Apprenticeship Learning (AL) (Abbeel & Ng, 2004). First, let us recall that an AL problem takes the following form (Ho & Ermon, 2016):

$$\min_{\pi} \max_{r \in \mathcal{R}} \mathbb{E}_{\mathbf{x}=(\mathbf{s},\mathbf{a}) \sim \mathcal{D}} \left[r(\mathbf{x})\right] - \mathbb{E}_{\mathbf{x}=(\mathbf{s},\mathbf{a}) \sim \rho_\pi} \left[r(\mathbf{x})\right], \tag{3}$$

where $\mathcal{R}$ is a set of reward functions. AL plays a min-max game between the policy $\pi$ and the reward function $r$. Intuitively, in the inner loop we would like to find a cost function such that the expert data's cummulative return is higher than that of the agent's, and their gap is maximized. Meanwhile, the policy $\pi$ tries to minimize this gap.

Now, let us consider how to reduce our objective to the AL formulation. For simplicity, we consider the case that we only have one negative sample. We notice that Equation 1 is then

$$\mathcal{L} = \mathbb{E}\left[-\log \frac{\exp \Phi(\mathbf{x}_0)^T \Phi(\mathbf{x}_p)}{\exp \Phi(\mathbf{x}_0)^T \Phi(\mathbf{x}_p) + \exp \Phi(\mathbf{x}_0)^T \Phi(\mathbf{x}_n)}\right]. \tag{4}$$

As suggested by (Khosla et al., 2020), we can apply the Taylor expansion trick to approximate this loss function with the following form:

$$\mathcal{L} \approx \mathbb{E}\left[\|\Phi(\mathbf{x}_0) - \Phi(\mathbf{x}_p)\|^2 - \|\Phi(\mathbf{x}_0) - \Phi(\mathbf{x}_n)\|^2\right]. \tag{5}$$

Note that we drop the constant terms and the scaling constant since they do not affect the optimization objective. Moreover, since $\Phi$ embeds the data points to the sphere, we have $\|\Phi(\mathbf{x})\|^2 = 1, \forall \mathbf{x}$. As a result, we can further expand each term above and have

$$\mathcal{L} = \mathbb{E}\left[\Phi(\mathbf{x}_0)^T \Phi(\mathbf{x}_n) - \Phi(\mathbf{x}_0)^T \Phi(\mathbf{x}_p)\right]. \tag{6}$$

Since the variables in this equation are independent from each other, we are minimizing

$$\mathcal{L} = \mathbb{E}_{\mathbf{x}_n \sim \rho_\pi}[\mathbb{E}_{\mathbf{x}_0 \sim \mathcal{D}}[\Phi(\mathbf{x}_0)]^T \Phi(\mathbf{x}_n)] - \mathbb{E}_{\mathbf{x}_p \sim \mathcal{D}}[\mathbb{E}_{\mathbf{x}_0 \sim \mathcal{D}}[\Phi(\mathbf{x}_0)]^T \Phi(\mathbf{x}_p)]. \tag{7}$$

Let the reward function $r_\theta(\mathbf{x}) = \mathbb{E}_{\mathbf{x}_0 \sim \mathcal{D}}[\Phi_\theta(\mathbf{x}_0)]^T \Phi_\theta(\mathbf{x})$ as we defined in Equation 2, then minimizing the Equation 7 is exactly doing the maximization of

$$\mathbb{E}_{\mathbf{x} \sim \mathcal{D}}\left[r_\theta(\mathbf{x})\right] - \mathbb{E}_{\mathbf{x} \sim \rho_\pi}\left[r_\theta(\mathbf{x})\right], \tag{8}$$

which is exactly the inner maximization loop of AL. Then optimizing the policy with respect to this $r_\theta$ is exactly the outer loop. Hence, our algorithm is reduced to AL.

## 4 EXPERIMENTS

In this section, we empirically evaluate PCIL on an extensive set of tasks from the DeepMind control suite (Tassa et al., 2018), a widely used benchmark for continuous control. Our experiments are designed to answer the following questions: (1) Can PCIL achieve expert performance, and how sample efficient is PCIL compared to state-of-the-art imitation learning algorithms? (2) How does the representation space of PCIL differ from that of the AIL methods? (3) How does our method perform when we use different representation learning methods and reward design?

### 4.1 EXPERIMENTAL SETUP

**Environments**  We experiment with 10 MuJoCo (Todorov et al., 2012) tasks provided by Deep-Mind Control Suite. The selected tasks cover various difficulty levels, ranging from simple control problems, such as the single degree of freedom cart pole, to complex high-dimensional tasks, such as the quadruped run. The episode length for all tasks is 1000 steps, where a per-step ground truth environment reward is in the unit interval $[0, 1]$. For each task, we train an expert policy using DrQ-v2 (Yarats et al., 2021) with the true environment reward function and use it to collect 10 demonstrations. We refer readers to Appendix A for the full task list and more details about the demonstrations.

**Training Details**  To update the encoder, we randomly sample 128 expert transitions and 128 agent transitions from a replay buffer. For arbitrary expert transition, any other expert transition is considered a positive sample, and all the agent transitions constitute the set of negative samples. We update the encoder by minimizing Equation 1 with respect to these samples. We use DrQ-v2 (Yarats et al., 2021) as the underlying RL algorithm to train the agent with the cosine similarity reward given in Equation 2. We use a budget of 2M environment steps for all the experiments. Further implementation details can be found in Appendix B.

**Baselines**  We compare PCIL to Behavioral Cloning (BC) and two major classes of imitation learning algorithms:

1. **Adversarial IRL:** We consider Discriminator-Actor-Critic (DAC) (Kostrikov et al., 2018), a state-of-the-art AIL method that employs an unbiased AIL reward function and performs off-policy training to reduce environmental interactions.

2. **Trajectory-matching IRL:** Primal Wasserstein Imitation Learning (PWIL) (Dadashi et al., 2020) and Sinkhorn Imitation Learning (SIL) (Papagiannis & Li, 2020) are two recently proposed trajectory-matching imitation learning methods. PWIL computes the reward based on an upper bound of Wasserstein distance. SIL computes the reward based on Sinkhorn distances (Cuturi, 2013).

To ensure a fair comparison, we implement all the baselines using the same RL algorithm. The implementation details of these algorithms are in the Appendix.

### 4.2 MAIN RESULTS

We show the performance curves of 6 tasks in Figure 3, which are averaged over three random seeds. More results on DeepMind control tasks are provided in Appendix D. We find that PCIL is able to outperform the existing methods on all of these tasks. It achieves near-expert performance within our online sample budget in all considered tasks except Hopper Hop. In terms of sample efficiency, i.e., the number of environment interactions required to solve a task, PCIL shows significant improvements over prior methods on five tasks: Cheetah Run, Finger Spin, Hopper Hop, Hopper Stand, and Quadruped Run. For the remaining tasks, PCIL achieves similar results compared with the state-of-the-art adversarial imitation learning method DAC. In particular, we notice that PCIL's performance gain is larger on more difficult tasks (e.g., Cheetah Run, Quadruped Run). On those easier tasks (e.g., Walker Stand, Walker Walk), the baselines are also able to achieve strong results.

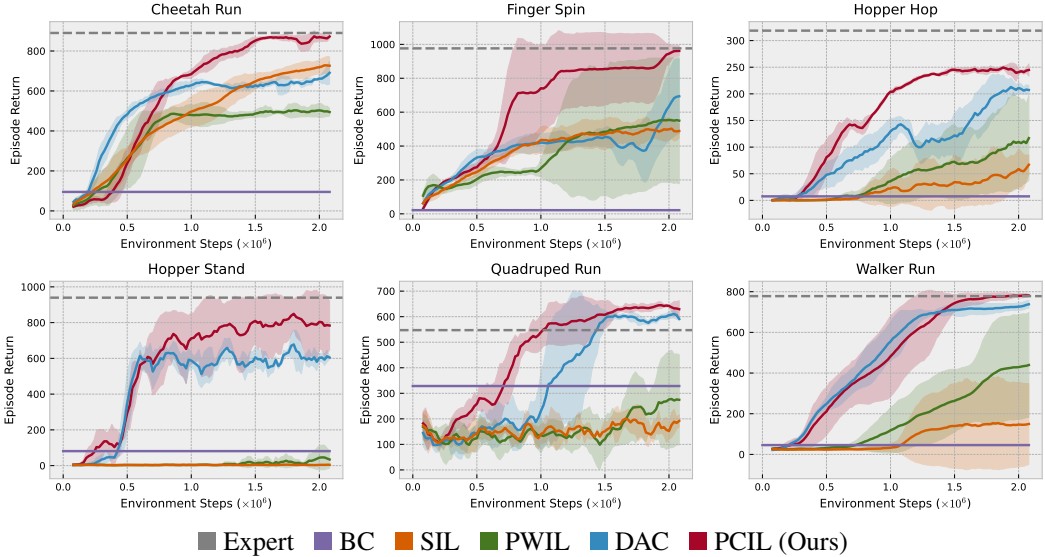

Figure 3: Comparisons of algorithms on 6 selected tasks. See Appendix D for more tasks. For every 20k environment steps, we perform 10-episode rollouts of the policy without exploration noise and report average episode returns over the 10 episodes. We plot the mean performance over 3 seeds together with the shaded regions, which represent 95% confidence intervals.

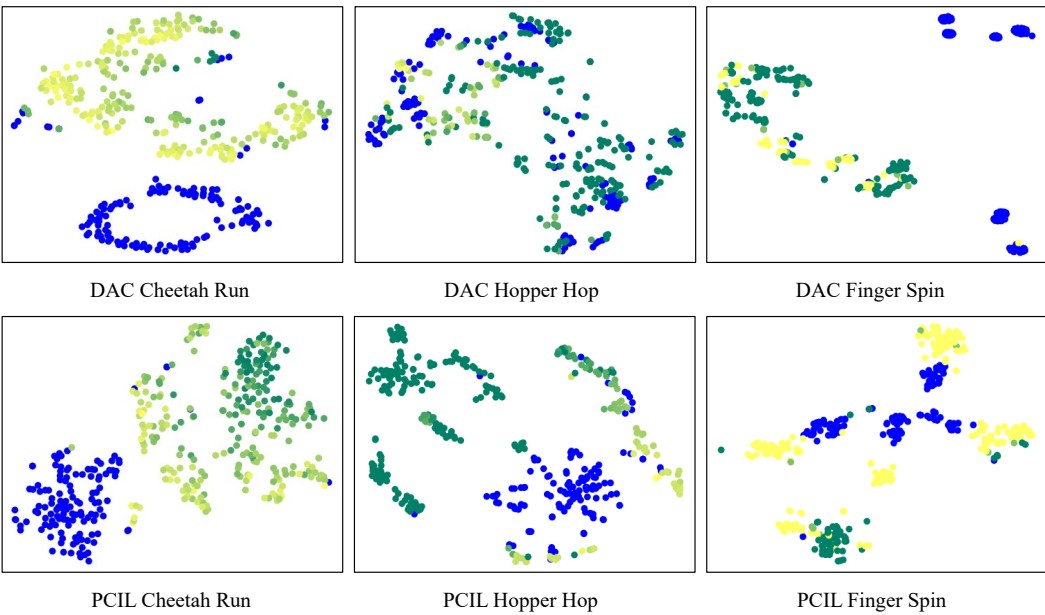

Figure 4: t-SNE visualization results for DAC (top panels) and our PCIL method (bottom panels). The blue color indicates the expert's transition. The lighter (yellow) color indicates agent transitions with higher real reward while the darker (green) indicates lower real reward.

## 4.3 ANALYSIS OF REPRESENTATION SPACE

We visualize the representation space of PCIL and DAC using t-SNE (Van der Maaten & Hinton, 2008) in Figure 4. For DAC, since there is no explicit representation learning in the discriminator, we treat the last hidden layer of its discriminator as the representation. We randomly sample 128 expert transitions and 256 agent transitions for visualization. For a fair comparison, all the transitions of

| Methods | Finger Spin | Walker Run | Hopper Stand | Hopper Hop |
|---|---|---|---|---|
| PCL + Sim-reward (Ours) | **964.2** ±**24.2** | **778.4** ±**14.1** | **771.2** ±**142.1** | **243.2** ±**20.1** |
| TCN + Sim-reward | **975.2** ±**45.1** | 180.5 ±11.4 | 2.0 ±1.2 | 6.8 ±0.2 |
| PCL + GAIL-reward | 2.5 ±2.4 | 18.5 ±6.5 | 1.2 ±1.8 | 0.1 ±0.1 |
| TCN + GAIL-reward | 35.4 ±19.2 | 19.8 ±4.2 | 1.5 ±0.3 | 17.3 ±4.6 |

Table 1: Ablation studies on the policy contrastive representation and similarity-based imitation reward. We report the average final returns on 4 selected tasks over 3 random seeds and standard deviations are given in error bars.

PCIL and DAC are selected from the same episode during the training process on the same task. We use color to indicate the real environment reward of the agent's transitions. The lighter (yellow) color indicates a higher reward for agent transition while the darker (green) indicates a lower reward agent transition. The blue color indicates the expert's transition.

We observe that in the representation space of PCIL, the expert transitions are concentrated in a cluster. Moreover, the distance between each agent transition to the cluster of expert transition is highly correlated with the real reward of that agent transition. This fact suggests that the contrastive objective of PCIL indeed induces a meaningful representation space here. On the contrary, the representation space of DAC is much less structured. The expert transitions are scattered throughout the representation space of DAC. Moreover, we identify that in the DAC's representation space, the agent's real ward does not correlate well with its distance to the expert transitions. These facts show that our method indeed learns a better representation space.

## 4.4 Ablation Studies

As described in our method, our method has two components: a policy contrastive representation for imitation (Section 3.2) and a similarity-based imitation reward (Section 3.3). In this part, we carry out ablation studies to analyze their effects. We first introduce our design choices as follows.

**PCIL Representation v.s. TCN Representation** We replace our proposed policy contrastive objective with other contrastive learning methods. For this purpose, we adopt the popular self-supervised representation learning method that leverages temporal information: Time-Contrastive Networks (TCN) (Sermanet et al., 2018). In this case, the positive samples are selected within a small window around the anchor sample, while the negative samples are selected from distant time steps in the same rollout trajectory. See Appendix C for implementation details.

**Similarity-based Reward v.s. GAIL-like Reward** We also ablate the similarity-based reward in PCIL by replacing the similarity-based imitation reward with a GAIL-like reward. Specifically, we train a linear binary classifier on the policy contrastive embedding space to distinguish expert or non-expert data. In this case, the embedding space is still trained by PCIL contrastive loss, and the GAIL reward's gradient is detached from the embedding network. We use the same reward predictor as other GAIL-style methods (Kostrikov et al., 2018), i.e. $\log(D(x)) - \log(1 - D(x))$.

**Analysis** By comparing rows 1 and 2 in Table 1, we find that the approach with TCN encoder does not work in three out of four environments. This is because the optimization goal of TCN is not to distinguish between expert and non-expert data. Thus the reward produced by comparing expert and non-expert data in the learned representation space is not necessarily meaningful. Note that the case in row 2 is no longer an adversarial IRL method. We also consider a case (row 4 in Table 1) where we use TCN and GAIL-like reward predictor, but the performance of this method is poor. Moreover, we observe that in the absence of the similarity-based imitation reward (compare rows 1 and 3 in Table 1), our method does not work. This is because our representation space has metric-space characteristics. As a result, we should use a distance-based metric to compute the reward. In conclusion, the two components of our method are necessary for achieving good performance.

## 5 RELATED WORK

### 5.1 IMITATION LEARNING

Imitation learning is a class of algorithms that enables a robot to acquire behaviors from a demonstration dataset. There are two classes of imitation learning algorithms: behavioral cloning (BC) and Inverse Reinforcement Learning (IRL) (Ng et al., 2000). BC is a simple supervised learning algorithm that directly fits the expert's action. However, some work suggests that it has some drawbacks: it suffers from covariate shift problem (Ross et al., 2011), and it is hard to learn from a demonstration dataset without expert actions (Torabi et al., 2018a). Instead, IRL (Abbeel & Ng, 2004) proposes to recover the underlying policy by estimating the underlying reward function and then maximizing the overall return with this reward. In particular, a recent branch of IRL is the AIL, which proposes to match agents' state-action distribution with experts via adversarial training. GAIL (Ho & Ermon, 2016) proposed a maximum entropy occupancy measure matching method which learns a discriminator to bypass the need to recover the expert's reward function. Later, several works proposed an improved version of the GAIL methods (Ghasemipour et al., 2020; Blondé et al., 2022; Baram et al., 2017; Kostrikov et al., 2018; Fu et al., 2017). AIRL (Fu et al., 2017) replaced the Shannon-Jensen divergence used in GAIL by Kullback-Leibler divergence to measure similarity between state-action pair distributions. Baram et al. (2017) bridges the GAIL framework to model-based reinforcement learning. DAC (Kostrikov et al., 2018) improved the sample efficiency by leveraging a replay buffer without importance sampling and dealing with the absorbing state problem. In contrast to these works, we focus on the representation of AIL's discriminator and reformulating AIL in a contrastive embedding space.

### 5.2 REPRESENTATION LEARNING FOR POLICY LEARNING

In this work, we propose a representation-learning-based approach to improve imitation learning. As imitation learning is a major class of policy learning algorithms, we review works that use self-supervised learning to improve policy learning in this part. Pioneer works (Mirowski et al., 2016; Jaderberg et al., 2016; Shelhamer et al., 2016; Lample & Chaplot, 2017) explore using auxiliary objectives (e.g., predict some property of the environment) as a self-supervision signal. Recent works employ more general self-supervised objectives (Oord et al., 2018). In particular, Srinivas et al. (2020) are based on contrastive representation learning. Sermanet et al. (2018) learn representation from multiview video using time contrastive learning. Some other methods also explore the use of self-supervised representation pretrained on environment data (Ha & Schmidhuber, 2018) or from real-world images (Xiao et al., 2022; Parisi et al., 2022; Nair et al., 2022). In imitation learning, Mandi et al. (2022) proposes to use contrastive learning for one-shot imitation learning in robotics. Chen et al. (2021) investigate the use of representation for imitation learning. However, their result suggests that self-supervised representation learning only provides a small improvement of imitation learning algorithms' performance. Our method differs from all these existing works by proposing to anchor on different policies and learn a discriminative self-supervised representation for imitation learning.

## 6 CONCLUSION

In this paper, we suggested a new approach to improve adversarial imitation learning algorithms: to learn a more meaningful, discriminative representation space for imitation. To this end, we proposed a new algorithm called PCIL. We conducted a theoretical analysis of our method and showed its connection to apprenticeship learning. We also conducted experiments on the DeepMind Control Suite and showed that PCIL could achieve state-of-the-art performance. Moreover, we used an ablation study to highlight its difference from the previous representation learning method. In the future, we will focus on further improving our loss function design. For example, can we anchor on the agent policies at different training stages? It will also be interesting to extend the proposed representation learning method in the relaxed setting of IL, like the scene where we can access both the reward and demonstration.

## 7 REPRODUCIBILITY

We implement our algorithm according to parameters and details described in Appendix B and Section 4.1. We will release our code and data.

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

## A  Environments

We use 10 continuous control tasks from the DeepMind control suite (Tassa et al., 2018). The summary for each task is provided in Table 2.

| Task | $\dim(\mathcal{S})$ | $\dim(\mathcal{A})$ |
|---|---|---|
| Finger Spin | 6 | 2 |
| Hopper Stand | 14 | 4 |
| Pendulum Swingup | 2 | 1 |
| Walker Stand | 18 | 6 |
| Walker Walk | 18 | 6 |
| Acrobot Swingup | 4 | 1 |
| Cheetah Run | 18 | 6 |
| Hopper Hop | 14 | 4 |
| Quadruped Run | 56 | 12 |
| Walker Run | 18 | 6 |

Table 2: A detailed description of each tasks used in our experiments.

**Demonstrations**   For each task, we train expert policies using DrQ-v2 (Yarats et al., 2021) on the actual environment rewards. We run 3 seeds and pick the seed that achieves the highest return. Then we use this expert policy to collect 10 demonstrations.

## B  Algorithm Details

### B.1  Implementation

**RL agent**   We use DrQ-v2 as the underlying RL algorithm. DrQ-v2 is an off-policy actor-critic algorithm for continuous control. The core of DrQ-v2 is Deep Deterministic Policy Gradient (DDPG) (Lillicrap et al., 2015) augmented with $n$-step returns. The critic is trained using clipped double Q-learning (Fujimoto et al., 2018) to reduce the overestimation bias in the target value. The deterministic actor is trained using deterministic policy gradients (DPG) (Silver et al., 2014). We also follows the setting of actor's and critic's neural network architectures in state-based DrQ-v2 Yarats et al. (2021).

**Contrastive encoder**   The contrastive encoder is implemented as a 4 layer MLP with hidden size [256, 256, 256]. The output dimension is 64. Following the architecture in Yarats et al. (2021), the contrastive encoder, the critic and the actor share the same encoder backbone. This shared encoder is trained with the gradient of the critic alone, which is also following the suggestion of Kostrikov et al. (2020); Yarats et al. (2021). The input of this shared encoder is state $s$ of a transition.

**Reward predictor**   Reward of the agent transition is computed according to Equation 2. Note that cosine similarity between expert data points is high due to the optimization goal described in Equation **??**. Thus, we randomly sample one expert transition from the expert replay buffer to compute agent reward. Empirically, we find that using the mean embedding of the expert data yields similar performance.

**Gradient penalty**   In order to make the algorithm more stable, we use the gradient penalty technique (Gulrajani et al., 2017) widely used in Wasserstein-GANs (Arjovsky et al., 2017). We make minor adjustments to accommodate our policy contrastive loss. GAIL-like methods usually constrain the gradient norm of the discriminator's output with respect to its input. While for PCIL, the contrastive encoder's output needs one more step. Specifically, the output embeddings are first used to calculate rewards following Equation 2, then we compute and penalize the gradient norm of the rewards. We use 10 as the weighting for the gradient penalty.

## B.2 HYPERPARAMETERS

Table 3 lists the hyperparameters that are used for all baseline methods and our method. Expert data ratio in PCIL means the ratio between expert data and batch size. A ratio of 0.5 means that half of the batch is expert data and the other half is the agent data. The contrastive learning usually needs a temperature scaling after computing the cos-similarity, before computing the exponential. For simplicity, we ignored it in the main text. In the experiment, we follow prior contrastive learning work He et al. (2020) and use a typical value of $0.07$ for the temperature.

| Methods | Parameter | Value |
|---------|-----------|-------|
| all methods | Replay buffer size | 500k |
| | Agent update frequency | 2 |
| | Optimizer | Adam |
| | Learning rate | 1e-4 |
| | Critic soft-update rate | 0.01 |
| | Random seed | 1,2,3 |
| | RL batch size | 128 |
| | Discriminator training batch size | 256 |
| | Hidden dim | 256 |
| PCIL | Expert data ratio | 0.5 |
| | Contrastive temperature | 0.07 |

Table 3: The hyperparameters of baseline methods and our method.

## C ABLATION STUDY IMPLEMENTATION DETAILS

### C.1 TCN REPRESENTATION

TCN encoder shares the same network architecture as the PCL encoder. During the training process in TCN encoder, for a random sampled anchor, we use data point adjacent to it as positive pair and another random sampled data point as negative pair. We also set contrastive temperature to $0.07$ and batch size to $256$, which is the same as PCL encoder.

### C.2 GAIL-LIKE REWARD

GAIL reward predictor can be seen as a simplified version of GAIL discriminator which has only one linear classifier layer. The reward predictor is trained independently to distinguish whether the input is from a expert data or non-expert data with a binary classification loss.

## D ADDITIONAL EXPERIMENTAL RESULTS

Figure 5 shows the performance of PCIL on the other 4 tasks from the DeepMind Control suite. We notice that the performance of some relatively easy tasks has saturated. All the baselines achieve expert performance on *Pendulum Swingup*. On *Walker Stand* and *Walker Walk*, PCIL is competitive with DAC, which already demonstrates impressive sample efficiency.

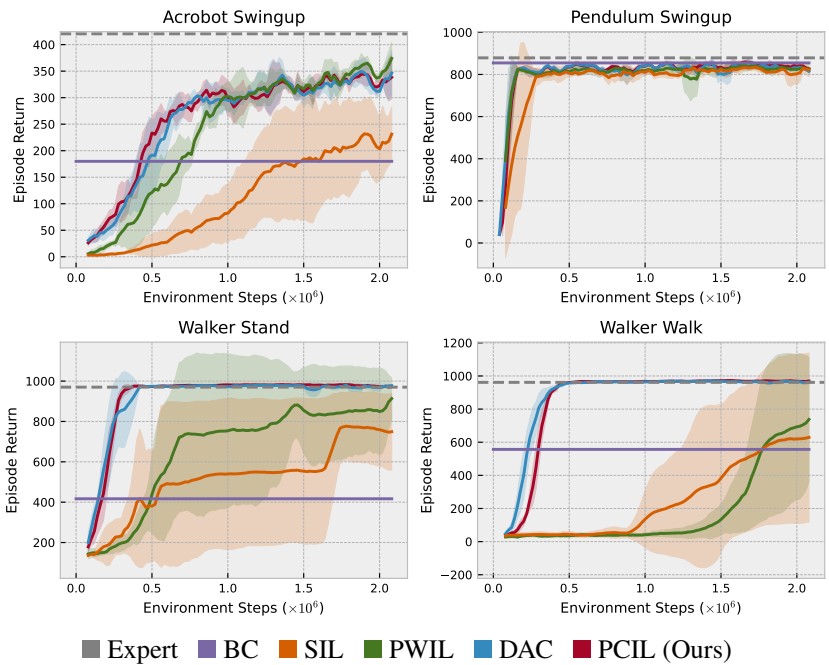

Figure 5: Comparisons of algorithms on the other 4 tasks.

