# OpenReview forum: "Policy Contrastive Imitation Learning"
_ICLR.cc/2023/Conference — Submitted to ICLR 2023_

### Official Review · Reviewer_igbW · 2022-10-13

**Confidence:** 3
**Clarity, Quality, Novelty And Reproducibility:** see above
**Correctness:** 2
**Technical Novelty And Significance:** 2
**Empirical Novelty And Significance:** 2
**Recommendation:** 5

**Strength And Weaknesses:**

**Strengths:**

I fully buy the motivation: GAIL is difficult to stabilize and representations may be unrobust. Also they may semantically not be meaningful.

Therefore, the idea to have an additional loss to improve representations makes sense, and contrastive learning may be a sensible way.

**Weaknesses**:

I think the paper is too premature in terms of both, writing and method.

Sec. 3.4 claims to deliver a theoretical analysis, but it stays pretty vague. Eq 3 seems central, but the derivation to get there from Eq. 2 is just informally described instead of adding at least 1-2 formal lines, especially since there are some many different $x$ version etc. where one can get lost in the expression.

Overall, intuitively I cannot follow the method's idea.
* I feel such a representation is not robust to distributional changes etc. I think it often happens that in one instance (say, for one initialization), a state-action pair is good (taken by the expert), while in another instance it is bad (taken by non-expert). I.e., whether a state is good or bad can heavily depend on initial state, random fluctuations, etc. E.g., from one starting position in a maze, the expert may never want to visit a certain state, while from another one, the state would lie on the shortest path. It's hard for me to make this intuitive counterargument specific, and maybe GAIL suffers from similar problems, but it does not seem to be a robust approach.
* It seems it does something similar as GAIL, but in a less principled way. But then why do we need to add it in the first place?
* Also I'm confused by relating the tuning of the representation, to tuning the policy. I think these are different things and I have a hard time seeing that Sec. 3.4 justifies this work from IRL/AL principles.

Some (minor) points re. writing:
* The overall method, i.e., GAIL + the new representation mechanism, is described too little; I feel it is just Fig. 2.
* "leanring" p3
* "markov" p3
* x is only informally introduced as state-action-pair. I strongly recommend to introduce all important notation in a formal, central way.
* what does "anchoring on policy" mean (p4)?

**Summary Of The Paper:**

In this paper, to address known issues of GAIL such as instabilities and non-robust representations, the authors propose an approach that additionally learns a representation in a contrastive way, by rewarding expert states to be nearby in the latent space, and expert-nonexpert states to be far apart. They contain some theoretical considerations as well as experiments.

**Summary Of The Review:**

The paper seeks to address an important problem, and contains some interesting ideas, but is too premature in terms of writing and analysis of the method.

---

### Official Review · Reviewer_dovc · 2022-10-14

**Confidence:** 3
**Correctness:** 2
**Technical Novelty And Significance:** 3
**Empirical Novelty And Significance:** 4
**Recommendation:** 5

**Clarity, Quality, Novelty And Reproducibility:**

**Clarity** -- The paper and figures are generally clear. Some of the wording and descriptions could be made more precise.

**Quality** -- The main experiments do a good job getting at the most important questions raised by the paper. As noted above, I'm a bit unsure about some of the conclusions drawn from Fig. 4.

**Novelty** -- This paper is novel, to the best of my knowledge. While it uses well-known ingredients, it combines them in an interesting way, and shows that alternative combinations work much worse (c.f. Table 1).

**Reproducibility** -- The paper does an OK job describing the experiment protocol. Some details (e.g., how actions are input to the models) are missing. No code is provided.

**Strength And Weaknesses:**

Strengths
* The proposed method is simple, and would be easy to apply on top of many different AIL algorithms.
* The experimental results are quite strong.
* Presentation. The text does a good job of clearly explaining the proposed method. The figures are generally easy to read.

Weaknesses
* I am a bit unsure about some of the math (see details below)
* Some of the writing is imprecise; some has grammatical errors (see below).

**Concern about the math**: My main concern about the math is that it's not entirely clear what the learned reward function (Eq. 2) actually corresponds to. Here's my attempt to figuring this out. Let $p(x, y) $ be the distribution over positive examples, and $p(x)p(y)$ be the corresponding marginal distributions. In our setting, we have $x = (s, a)$ and $p(x, y) = \rho_E(s, a)\rho_E(s', a')$. For simplicity, let's assume that we just use 1 negative example. Then the marginal distributions are $p(x) = \rho_E(s, a)$ and $p(y) = \frac{1}{2}\rho_E(s', a') + \frac{1}{2} \rho_\pi(s', a')$. The optimal critic $f(x, y)$ satisfies $e^{f(x, y)} = \frac{p(x, y)}{p(x)p(y)c(x)}$, where $c(x)$ is an arbitrary positive function [1, 2]. In our setting, this means we have $e^{f(s, a, s', a')} = \frac{\rho_E(s, a)\rho_E(s', a')}{\rho_E(s, a) \frac{1}{2}(\rho_E(s', a') + \rho_\pi(s', a'))c(s, a)}$. The $\rho_E(s, a)$ terms in the numerator and denominator cancel, so we have $e^{f(s, a, s', a')} = \frac{\rho_E(s', a')}{\frac{1}{2}(\rho_E(s', a') + \rho_\pi(s', a'))c(s, a)}$. Written in terms of the representations, we have $\Phi(s, a)^T \Phi(s', a') = \log \rho_E(s', a') - \log \left(\frac{1}{2}(\rho_E(s', a') + \rho_\pi(s', a')) \right) - \log c(s, a)$. The only term that depends on $s, a$ is the $\log c$ term, which is entirely arbitrary. If this is correct, it suggests that the proposed method maximizes a completely arbitrary reward function, not minimizing any sort of divergence measure. I suspect there's a bug somewhere in my math above, but hopefully it will be helpful in analyzing what the learned reward function actually is.

**Other questions and concerns**:
* "fragility of the discriminator leads to ..." -- How do we know that the problem is with the discriminator, and not with some other part of the learning algorithm? Consider adding a citation/evidence.
* Fig 1 -- I found this figure a bit misleading. The claim that "the embedding space is not required to be semantically meaningful" would seem to suggest that classification doesn't lead to good representation spaces. Not only is this likely untrue (e.g., ImageNet does learn semantically meaningful features), but also the paper later claims that the infoNCE loss (an instance of classification) _does_ yield good representations. The claim that "our method enforces compactness" could be easily imposed on existing methods by normalizing their representations, without requiring any changes to the objectives.
* "As a result, the model may not distinguish between good and bad states effectively." -- I didn't understand this point. It seems like existing AIL methods should be able to distinguish this, if given enough data and enough model capacity. Why do they fail?
* Sec 3.4 -- I found this section a bit confusing, partially because it was unclear exactly what the aims were. It might be helpful to exactly write out the apprenticeship learning IRL reward (in the general case), and then show that the proposed method corresponds to a special case. An ungenerous reading of this section is that it just says that the proposed method is maximizing a linear reward function, which is trivially true for any AIL method (the reward is a linear function of the last-layer features).
* Sec 4.2 -- One potential confounder in the experiments is the scale of the learned reward functions. It might be good to study this. E.g., if the scale of the DAC rewards is forced to be similar to the PCIL rewards, does DAC perform better?
* Sec. 4.3 -- I'm not sure the evidence provided in Fig 4 supports the claims in this section. To me, the DAC representations look like they *better* separate expert vs non-expert data, and that the distance from the expert data is *more* correlated with the rewards (as compared with the PCIL representations).


**Minor writing comments**
* "to encourage imitation learning" -- Unclear what this means. Taken literally, it seems like an imitation learning method wouldn't need extra "encouragement" to do imitation learning.
* "One recent class" -- 2016 isn't particularly recent.
* "error issue (Ross et al" -- This makes it seem like a 2011 paper proved a result about a 2016 paper, which would be impossible.
* "superior performance" -- Cite.
* "without expert actions" -- Cite.
* "etc" -- Avoiding using "etc" in technical writing.
* "semantically meaningful signals" -- Unclear what this means
* "Instead of using a discriminator ... we propose to train a discriminator representation" -- This seems like a type error, comparing a discriminator to a representation.
* "the binary classifier's representation space does not necessarily satisfy our objective" -- I didn't understand this sentence.
* "Suite Tassa" -- Use \citep instead of \citet here.
* "very non-robust" -- Robust w.r.t. what criterion? Add a citation. Stylistically, I'd remove the "very," too.
* "Researchers have found" -- Cite.
* "people have found" --> "prior work~\citep{...} has found"
* Fig 2 -- This figure is great for explaining the method!
* "definition of similarity in the previous works" -- What is this definition? It'd be helpful to include the definition here.
* "Though combining ..." -- I didn't understand the strawman construction here.
* "pair-wise distance constraint" -- What is this constraint?
* "$\Phi(x_0)^T\Phi(x) > t$" -- How was this derived?
* "the binary classification induced latent space might not satisfy our constraint." -- I didn't understand this point.
* "Theoretical Analysis" -- I'd recommend renaming this "Connections with Feature Matching" because the section doesn't include any formal theorems/lemmas.
* Eq. 3 -- Perhaps cite [3] or similar prior work that uses this objective
* "we will find that the" -- How is this proven?
* "three random seeds" -- I'd highly recommend running at least 2 additional seeds.
* Fig 3 -- increase the size of the xlabel and ylabel fonts.
* Fig 4 -- Add a takeaway to the caption, so the reader knows what they are supposed to learn from this plot.
* "distant time steps in the same trajectory" -- I'd recommend highlighting this, using it to motivate why sampling positive examples from different trajectories is better than sampling positive examples from the same trajectory.
* Table 1 -- In column 1, both the first and second rows should be highlighted. The caption should specify the number of random seeds and indicate what the error bars correspond to. Table captions should appear before the table.
* "representation space has metric-space characteristics" -- Every representation space forms a metric space.
* I'd highly recommend running a grammar checker on the paper (e.g., copy+paste into a Google Doc and run the grammar checker)
* Conclusion -- This is well written.
* "Russel, et al." -- Why are the rest of the authors not listed?
* Table 2 -- Are these image-based tasks or state-based tasks? Captions should appear above the table.
* Gradient penalty -- I'd recommend showing an equation or pseudocode to help clarify this section.
* "we ignored it in the main text" -- Please add this detail back to the main text.

[1] https://arxiv.org/pdf/1809.01812.pdf

[2] http://proceedings.mlr.press/v97/poole19a/poole19a.pdf

[3] https://arxiv.org/abs/2106.04156

**Summary Of The Paper:**

The main idea in this paper is to do adversarial imitation learning (AIL) with a discriminator trained via the infoNCE loss, rather than the binary cross-entropy loss. The paper argues that this choice of loss function will result in "smoother" and more "semantically meaningful" reward signals for training the policy. Experimentally, the paper shows that the proposed method improves performance (often significantly) compared with DAC and other imitation learning baselines on the challenging image-based DM Control benchmark.

**Summary Of The Review:**

Overall, the empirical results from the paper look strong, and I like the simplicity of the proposed method. I am a bit concerned that the math behind the proposed method doesn't check out. If this math is clarified, and if the writing clarity is improved (see detailed comments above), I will advocate for accepting the paper.

---

### Official Review · Reviewer_1ZsP · 2022-10-22

**Confidence:** 4
**Correctness:** 4
**Technical Novelty And Significance:** 4
**Empirical Novelty And Significance:** 3
**Recommendation:** 8

**Clarity, Quality, Novelty And Reproducibility:**

* Clarity: The paper is very easy-to-follow. The figures are helpful for understanding.
* Quality: The idea is technically sound thanks to the theoretical result. The results and analysis are strong and convincing.
* Novelty: Although contrastive learning has been proposed in the context of supervised/unsupervised learning, the application of the idea to imitation learning in RL is novel. The theoretical justification using the apprenticeship learning framework is novel and interesting,
* Reproducibility: The paper provided the details of architectures and hyperparameters in the appendix and promised to release the code.

**Strength And Weaknesses:**

[Strength]

* The idea of using contrastive learning and similarity-based reward is novel and interesting.
* The theoretical result is also novel and nicely justifies the proposed method.
* The performance on various MuJoCo tasks looks strong.
* The analysis of the learned representation makes the method more convincing.
* The paper is well-written.

[Weakness]

* Lack of robustness analysis. In the introduction, the paper motivates the problem by pointing out how fragile the discriminator is in existing adversarial imitation learning algorithms. However, there is no experiment demonstrating the robustness of the proposed method. It would be even more convincing if the paper showed that the proposed method is much more robust to the choice of the hyperparameters and architectures of the proposed discriminator.

**Summary Of The Paper:**

This paper proposes a new adversarial imitation learning method, where a discriminator is trained to learn a representation such that the distance between expert trajectories are shorter than the distance between agent-expert trajectories. This representation is then used to calculate an imitation reward for the agent to encourage it to generate trajectories that are more expert-like. The paper shows that this representation amounts to feature matching between the expert and the agent. The results on MuJoCo tasks show that the proposed method significantly outperforms the state-of-the-art adversarial imitation learning methods.

**Summary Of The Review:**

This paper proposes an interesting and novel idea with a nice theoretical backup. The results look solid with a minor caveat that I pointed out above. Thus, I believe that this paper is interesting enough to be presented at ICLR.

---

### Author Response · Authors · 2022-11-22
**More Theoretical Analysis (Part 1/2)**


Dear readers, we have updated the theoretical result. **We can further show that we are minimizing the total variation divergence between the expert and policy distribution.** The analysis is as follows.

At the last of Section 3.4, we have shown that our learning process is playing a min-max game:
$$
    \min_{\pi}\max_{r_\theta}  E_{x\sim \mathcal{D}} r_\theta(x) -  E_{x\sim \rho^\pi} r_\theta(x),
$$

where $r_\theta(x) = (E_{x'\sim\mathcal{D}}\Phi_\theta(x')dx')^T\Phi_\theta(x)$, and $\mathcal{D}$ is the expert dataset. We may as well regart $\mathcal{D}$ as $\rho^E$ when the dataset size is large.

Define the inner objective as
$$
D_{cont}(\rho^E||\rho^\pi) = \max_{r_\theta} [E_{x\sim \rho^E} r_\theta(x) - E_{x\sim \rho^\pi} r_\theta(x)],
$$
then $\pi$ is doing the following minimization problem:
$$
\min_{\pi} D_{cont}(\rho^E||\rho^\pi).
$$
In this part, we will show that

> **Theorem 1**
$$
    0.25 D_{TV}(\rho^E||\rho^\pi)  \leq D_{cont}(\rho^E||\rho^\pi) \leq 2.0 D_{TV}(\rho^E||\rho^\pi).
$$
Here, $D_{TV}(\rho^E||\rho^\pi) = \frac{1}{2}\int_\mathcal{X} |\rho^E(x) - \rho^\pi(x)|dx$ is the total variation divergence. Therefore, our objective is a proxy of $D_{TV}(\rho^E||\rho^\pi)$.

This theorem states that $D_{cont}(\rho^E||\rho^\pi)$ is actually a diveregence and is controlled by the total variance divergence. As a result, we will find that the outer optimization over $\pi$ can be regarded as minimizing the $D_{TV}(\rho^E||\rho^\pi)$.


**Proof of Theorem 1**

**Step 1** We first turn the inner optimization to a simple form. Let $v_\theta = E_{x'\sim\rho^E}\Phi_\theta(x')dx'$, then we find a corresponding orthonormal matrix $M_\theta$ such that $M_\theta v_\theta  = [\Vert v_\theta\Vert, 0, 0, 0, ..., 0]^T$. As a result,
$$r_\theta = (E_{x'\sim\rho^E}\Phi_\theta(x')dx')^T\Phi_\theta(x) = (M_\theta E_{x'\sim\rho^E}\Phi_\theta(x')dx')^T M_\theta\Phi_\theta(x) = \Vert v_\theta\Vert [M_\theta\Phi_\theta(x)]_1.
$$
If we define $g_\theta = [M_\theta\Phi_\theta(x)]_1: \mathcal{X}\to [-1, 1]$, then the inner optimimzation problem on $r_\theta$ can be written as:

> $ \mathsf{InnerOPT}$
Maximize (w.r.t. $\theta$, $\alpha$)
$$
    \alpha \int_\mathcal{X} g_\theta(x) [\rho^E(x) - \rho^\pi(x)] dx,
$$
subject to
$$
    \alpha = \left|\int_\mathcal{X} g_\theta(x) \rho^E(x)dx\right|.
$$

Note that the constraint is derived from the following fact:
$$
\alpha = \Vert v_\theta\Vert = \Vert M_\theta v_\theta\Vert = \Vert  E_{x'\sim\rho^E}M_\theta\Phi_\theta(x')\Vert = \left|\int_\mathcal{X} g_\theta(x) \rho^E(x)dx\right|.
$$


**Step 2** In this step, we show that $0.25 D_{TV}(\rho^E||\rho^\pi)  \leq D_{cont}(\rho^E||\rho^\pi)$. We do this by constructing a $g_\theta$, and evaluate the corresponding objective function. First, let $S = \\{x\in\mathcal{X}: \rho^E(x)\geq\rho^\pi (x)\\}$. Then, we split the discussion into two cases.

**Case 2.1** $\mu^E(S)\geq \mu^E(S^c)$.

We define
$$
    g_\theta(x) = \begin{cases}
1 &\text{$x\in S$},\\
-\beta &\text{$x \in S^c$}
\end{cases} .
$$
where $\beta\in [0, \mu^E(S)]$ is a scalar. Then, we know that
$$
    \alpha = |\mu^E(S) - \beta \mu^E(S^c)| = |\mu^E(S) - \beta (1 - \mu^E(S))| = |(1+\beta)\mu^E(S) - \beta| = (1+\beta)\mu^E(S) - \beta.
$$
Meanwhile, we know
$$
\int_\mathcal{X} g_\theta(x) [\rho^E(x) - \rho^\pi(x)] dx = \int_S g_\theta(x) [\rho^E(x) - \rho^\pi(x)] dx + \int_{S^c} g_\theta(x) [\rho^E(x) - \rho^\pi(x)] dx
$$
and
$$
RHS = \int_S |\rho^E(x) - \rho^\pi(x)| dx + \beta\int_{S^c} |\rho^E(x) - \rho^\pi(x)| dx \geq \beta \int_\mathcal{X} |\rho^E(x) - \rho^\pi(x)|dx
$$
Therefore, we have
$$
 D_{cont}(\rho^E||\rho^\pi) \geq \alpha \int_\mathcal{X} g_\theta(x) [\rho^E(x) - \rho^\pi(x)] dx \geq ((1+\beta)\mu^E(S) - \beta)\beta \int_\mathcal{X}|\rho^E(x) - \rho^\pi(x)|dx = 2((1+\beta)\mu^E(S)-\beta)\beta D_{TV}(\rho^E||\rho^\pi).
$$
Since $\mu^E(S)\geq \mu^E(S^c)$ and $ \mu^E(S) + \mu^E(S^c) = 1$, we have $\mu^E(S)\geq 0.5$. Then,
$$
RHS = 2((1+\beta)\mu^E(S)-\beta)\beta D_{TV}(\rho^E||\rho^\pi) \geq (1-\beta) \beta D_{TV}(\rho^E||\rho^\pi).
$$
In particular, if we pick $\beta = 0.5$, then the right hand side is $0.25D_{TV}(\rho^E||\rho^\pi)$, so in this case,
$$
D_{cont}(\rho^E||\rho^\pi) \geq 0.25D_{TV}(\rho^E||\rho^\pi).
$$
(Note that the $g_\theta$ defined above can be realized by some $\Phi_\theta$, see the end of the post.)

---

> ### Author Response · Authors · 2022-11-22
> **More Theoretical Analysis (Part 2/2)**
>
> (Cont'd)
> **Case 2.2** $\mu^E(S) < \mu^E(S^c)$.
> This can be done in a similar way.
>
> We define
> $$
>     g_\theta(x) = \begin{cases}
> \beta &\text{$x\in S$},\\
> -1 &\text{$x \in S^c$}
> \end{cases} .
> $$
> where $\beta\in [0, 0.5]$ is a scalar. Then, we know that
> $$
>     \alpha = |\beta\mu^E(S) - \mu^E(S^c)| = |\beta\mu^E(S) - (1 - \mu^E(S))| = |(1+\beta)\mu^E(S) - 1| = 1 - (1+\beta)\mu^E(S).
> $$
> Meanwhile, we know
> $$
> \int_\mathcal{X} g_\theta(x) [\rho^E(x) - \rho^\pi(x)] dx = \int_S g_\theta(x) [\rho^E(x) - \rho^\pi(x)] dx + \int_{S^c} g_\theta(x) [\rho^E(x) - \rho^\pi(x)] dx
> $$
> and
> $$
> RHS = \beta\int_S |\rho^E(x) - \rho^\pi(x)| dx + \int_{S^c} |\rho^E(x) - \rho^\pi(x)| dx \geq \beta \int_\mathcal{X} |\rho^E(x) - \rho^\pi(x)|dx
> $$
> Therefore, we have
> $$
>  D_{cont}(\rho^E||\rho^\pi) \geq \alpha \int_\mathcal{X} g_\theta(x) [\rho^E(x) - \rho^\pi(x)] dx \geq (1 - (1+\beta)\mu^E(S))\beta \int_\mathcal{X}|\rho^E(x) - \rho^\pi(x)|dx = 2(1 - (1+\beta)\mu^E(S))\beta D_{TV}(\rho^E||\rho^\pi).
> $$
> Since $\mu^E(S^c) > \mu^E(S)$ and $ \mu^E(S) + \mu^E(S^c) = 1$, we have $\mu^E(S) < 0.5$. Then,
> $$
> RHS = 2(1 - (1+\beta)\mu^E(S))\beta D_{TV}(\rho^E||\rho^\pi) \geq (1-\beta) \beta D_{TV}(\rho^E||\rho^\pi).
> $$
> In particular, if we pick $\beta = 0.5$, then the right hand side is $0.25D_{TV}(\rho^E||\rho^\pi)$, so in this case,
> $$
> D_{cont}(\rho^E||\rho^\pi) \geq 0.25D_{TV}(\rho^E||\rho^\pi).
> $$
> Putting Case 1 and 2 together, we can conclude that
> $$
> D_{cont}(\rho^E||\rho^\pi) \geq 0.25D_{TV}(\rho^E||\rho^\pi).
> $$
>
> **Step 3** Finally, we show that
> $$
> D_{cont}(\rho^E||\rho^\pi) < 2.0D_{TV}(\rho^E||\rho^\pi).
> $$
> This is actually quite straightforward. Note that
> $$
>     \alpha = \left|\int_\mathcal{X} g_\theta(x) \rho^E(x)dx\right| \leq \int_\mathcal{X} |g_\theta(x) \rho^E(x)| dx = \int_\mathcal{X} |g_\theta(x)| \rho^E(x)dx\leq \int_\mathcal{X} \rho^E(x)dx = 1,
> $$
> we have
> $$
>     \alpha \int_\mathcal{X} g_\theta(x) [\rho^E(x) - \rho^\pi(x)] dx \leq |\alpha| |\int_\mathcal{X} g_\theta(x) [\rho^E(x) - \rho^\pi(x)] dx| \leq |\int_\mathcal{X} g_\theta(x) [\rho^E(x) - \rho^\pi(x)] dx| \\ \leq \int_\mathcal{X} |g_\theta(x)||\rho^E(x) - \rho^\pi(x)| dx \leq \int_\mathcal{X} |\rho^E(x) - \rho^\pi(x)| dx = 2D_{TV}(\rho^E||\rho^\pi).
> $$
> This then completes the proof.
>
> Thank you.
> Authors
>
> -------------------------------------------------------------------
>
> Note: One can easily see that the $g_\theta$ defined in Step 2 can be realized by some $\Phi_\theta$. Take Case2.1 as an example, we can find out a $\Phi_\theta:\mathcal{X}\to\mathbb{S}$ to realize this $g_\theta$ as follows. First, let us split $S^c$ into two disjoint parts $A$, $B$ such that $A\cup B = S^c, A\cap B=\emptyset, \mu^E(A)=\mu^E(B)$ (this can be done for the continuous space considered in this paper). Then let us define
> $$
>     \Phi_\theta(x) = \begin{cases}
>                   (1, 0) &\text{$x\in S$},\\
>                   (-\frac{1}{2}, \frac{\sqrt{3}}{2}) &\text{$x \in A$}, \\
>                   (-\frac{1}{2}, -\frac{\sqrt{3}}{2}) &\text{$x \in B$}
>                   \end{cases} .
> $$
> Then we will see that $E_{x\sim\rho^E}(\Phi_\theta(x)) = (\mu^E(S)-\frac{1}{2}\mu^E(S^c), 0)$. Hence, the corresponding orthonormal matrix $M_\theta = I$ is the identity matrix. So, $g_\theta = [M_\theta\Phi_\theta(x)]_1$ is what we exactly defined at the end of Case2.1.

---

### Decision · Program_Chairs · 2023-01-20

**Decision:**

Reject

**Justification For Why Not Higher Score:**

Overall, the core innovation here is potentially clever, but the empirical results are not sufficiently strong on their own to be convincing given that 2/3 reviewers have concerns about the theoretical results.

**Justification For Why Not Lower Score:**

N/A

**Metareview: Summary, Strengths And Weaknesses:**

The authors propose policy contrastive imitation learning as an alternative to generative adversarial imitation learning. The authors assert that the primary shortcoming of GAIL-style approaches is that the discriminator is essentially trained to be a binary classifier. By using a contrastive loss that pulls expert-samples together and pushes non-expert samples away from the expert samples, the authors claim they produce a discriminator that better captures how expert behavior differs from sub-optimal and random behavior.

The reviews were a bit split, with one reviewer being very impressed by the paper, while two other reviewers rated the paper just below the acceptance threshold.  While all reviewers appreciated the simple and clear innovation and found the empirical results reasonable, the two negative reviewers were not convinced the theoretical support was sound.

**Summary Of Ac-Reviewer Meeting:**

We (the AC and reviewers) had a meeting to discuss the paper.  Unfortunately two of the reviewers were sick during the review period and ultimately reviewer igbW was not able to participate in the meeting.

At the meeting, reviewer dovc reiterated that the revision to the theoretical results notwithstanding, they identified what they believe to be a fundamental issues with the theory.  On the other hand, reviewer 1ZsP continues to feel the contribution is worthwhile.  However, 1ZsP had not taken a very close look at the specific concerns raised by dovc.  Specifically, dovc pointed to their concern in the original theoretical analysis, and the authors didn't really directly respond to the identified issue, instead proposing a different theoretical result in their rebuttal.  This alternative analysis didn't satisfy the original concern.  Unfortunately, it seemed that the authors may have not fully understood the concern raised by dovc, and though I encouraged dovc to further expand on their concern in the discussion, they did not.  Overall, the author revisions to the theory and dovc's concerns mostly talk a bit past one another, which is somewhat unfortunate as it made it harder to resolve.